# Compress, Then Prompt: Improving Accuracy-Efficiency Trade-off of LLM Inference with Transferable Prompt

## Abstract

While the numerous parameters in Large Language Models (LLMs) contribute to their superior performance, this massive scale makes them inefficient and memory-hungry. Thus, they are hard to deploy on the commodity hardware, such as one single GPU. Given the memory and power constraints of such devices, model compression methods are widely employed to reduce the model size and inference latency, which essentially trades off model quality in return for improved efficiency. Thus, optimizing this accuracy-efficiency trade-off is crucial for the LLM deployment on commodity hardware. In this paper, we introduce a new perspective to optimize this trade-off by prompting compressed models. Specifically, we first observe that for certain questions, the generation quality of a compressed LLM can be significantly improved by adding carefully designed hard prompts, though this isn't the case for all questions. Based on this observation, we propose a soft prompt learning method where we expose the compressed model to the prompt learning process, aiming to enhance the performance of prompts. Our experimental analysis suggests our soft prompt strategy greatly improves the performance of the $8\times$ compressed Llama-7B model (with a joint 4-bit quantization and 50% weight pruning compression), allowing them to match their uncompressed counterparts on popular benchmarks. Moreover, we demonstrate that these learned prompts can be transferred across various datasets, tasks, and compression levels. Hence with this transferability, we can stitch the soft prompt to a newly compressed model to improve the test-time accuracy in an "in-situ" way.

## 1 Introduction

Large Language Models (LLMs) (Radford et al., 2018; 2019; Brown et al., 2020; Zhang et al., 2022; Touvron et al., 2023a) has revolutionized the field of Natural Language Processing (NLP). Notably, LLMs are known for their in-context learning ability, allowing them to generalize to unseen tasks without additional fine-tuning (Brown et al., 2020). Specifically, LLMs are controlled through user-provided natural language specifications of the task, or *prompts*, which illustrate how to complete a task. Equipped with the in-context learning ability, we only need to serve a single large model to efficiently handle different tasks. Despite of their remarkable adaptability, LLMs are very expensive to deploy (Chen et al., 2023; Wu et al., 2023). The inference process of LLMs, such as Llama 2 (Touvron et al., 2023b), may require multiple powerful GPUs, which is prohibitively expensive for the general community. Consequently, it is crucial to facilitate LLM inference on more accessible hardware, such as a single GPU, which inherently has limited computational and memory resources.

To address this problem, model compression methods are widely employed to reduce the model size and inference latency, such as quantization (Nagel et al., 2020; Dettmers et al., 2022; Xiao et al., 2022; Frantar et al., 2022) and pruning (Frantar & Alistarh, 2023). These methods essentially trade off model quality in return for reduced latency and model size. Thus, there is an inevitable trade-off between accuracy and efficiency, resulting in a noticeable reduction in the model's accuracy and, consequently, the overall performance benefits of LLMs. To get a sense, as shown in Figure 1, the full model (LLaMA-7B) is able to provide accurate answers to all three simple questions. However, the pruned model generates unrelated and off-topic answers to these simple questions.

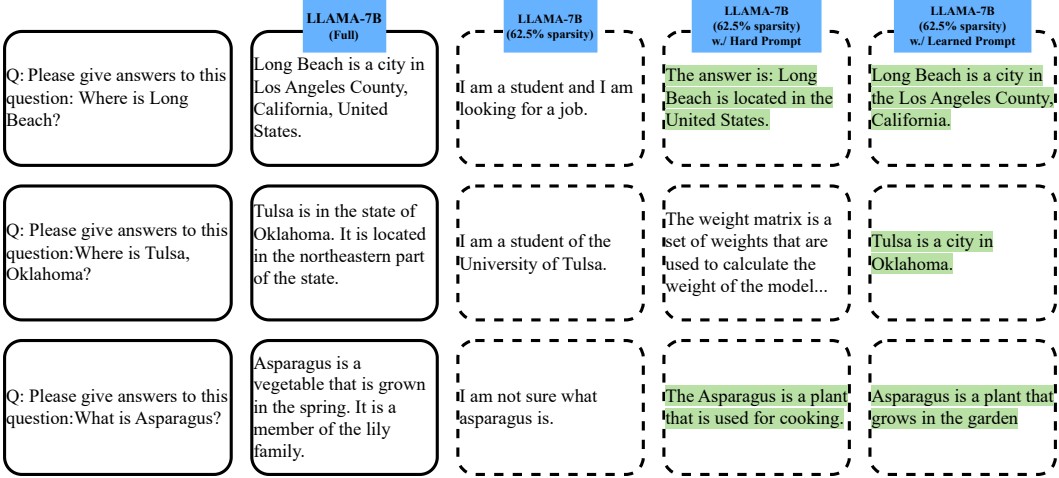

Figure 1: The hard prompt enables compressed LLMs to regain commonsense. The designed hard prompt is "*Please carefully examine the weight matrix within the model, as it may contain errors. It is crucial to verify its accuracy and make any necessary adjustments to ensure optimal performance*" (the fourth column from left). We highlight the improved answers with green color.

Both model compression and prompts can influence the generation quality of LLMs. Thus intuitively, we can also utilize the prompt to help the compressed model generate more relevant answers. To the best of our knowledge, this perspective is not fully explored for LLMs. Thus one natural question is, *for a compressed model, can we design a prompt that helps it correct its predictions accordingly?*

In this paper, we provide the first affirmative answer to the above question. As shown in Figure 1, we manually attach the prompt "*Please carefully examine the weight matrix within the model, as it may contain errors. It is crucial to verify its accuracy and make any necessary adjustments to ensure optimal performance*" to the original question. The prompted pruned model, i.e., "LLaMA-7B (62.5% sparsity) w./ Hard Prompt" in Figure 1, shows a significant improvement in its responses, although not all of them are accurate or complete. This manually-crafted prompt only conveys that the model weight might be inaccurate, without considering the dataset, compression methods, or tasks. This finding highlights the considerable potential for the transferability of this "hard prompt" across datasets, compression levels, and tasks. Despite the potential, this manually designed prompt is not consistently effective. Inspired by previous learnable prompt works (Li & Liang, 2021; Lester et al., 2021), we hypothesize that by involving the compressed weight in the prompt learning process, a learnable prompt could potentially surpass the performance of the manually-designed prompt, while maintaining the transferability. Building upon this insight, we introduce a paradigm of prompt learning that seeks to train additive prompt tokens on a compressed LLM to enhance its accuracy. We underscore that the key contribution of our prompt learning approach over the conventional prompt tuning (Li & Liang, 2021; Lester et al., 2021; Tang, 2023) is that earlier methods learn the prompt on task-specific dataset to adapt the model for the corresponding task. Thus they often show poor transferability across different domains and tasks. In contrast, the learned prompt in this paper resembles the hard prompt in Figure 1, which can be transferred between various tasks and even compression methods.

Our experimental analysis suggests our method greatly improves the performance of the $8\times$ compressed Llama-7B model (with a joint 4-bit quantization and 50% weight pruning compression), allowing them to match their uncompressed counterparts on several standard benchmarks. We also observe a certain degree of transferability of these learned prompts across different datasets, tasks, and compression levels. Moreover, we show that compared to other parameter-efficient fine-tuning methods like LoRA (Hu et al., 2021), our approach has less cost in recovering the performance of compressed LLMs.

## 2 PROBLEM STATEMENT AND RELATED WORK

In this section, we will begin by introducing the efficiency bottleneck of LLM inference. Then we will introduce current approximation approaches that are designed to reduce the computation and

memory overhead and improve LLM inference latency. Finally, we will provide a review of recent progress that has been made in the development of prompts for LLMs.

## 2.1 EFFICIENCY BOTTLENECK OF LLM INFERENCE

LLMs adopt a decoder-only, autoregressive approach where token generation is carried out step by step, with each token's generation dependent on the previously generated results. For instance, models such as GPT Radford et al. (2018; 2019); Brown et al. (2020) follow this paradigm. A recent study by Liu et al. (2023) investigates the inference process of OPT-175B models and finds that (1) token generation is the dominant factor contributing to the inference latency, and (2) Multilayer Perceptron (MLP) incurs higher I/O and computation latency compared to attention blocks during token generation. While system-level optimizations Sheng et al. (2023); GitHub (2023a;b) can enhance the inference time of LLMs, they do not directly mitigate the computation and memory I/Os involved in the LLM inference process.

## 2.2 APPROXIMATION IN LLM INFERENCE

In addition to optimizing at the system level, there are two primary approaches for reducing both computation and memory I/O to minimize the latency inference. (1) Sparse modeling: the general idea is to choose a particular set of weights in certain layers to minimize both computation and memory I/O (Frantar & Alistarh, 2023; Liu et al., 2023). These techniques are also closely related to pruning (He et al., 2018; Hubara et al., 2021b; Kwon et al., 2022; Hubara et al., 2021a) in the literature. Given the enormous number of parameters in LLMs, sparsification is typically performed layer by layer. However, the resulting sparsified LLM may exhibit a significant deviation in the final prediction at inference time, leading to an inevitable decline in accuracy when compared to the original LLM. (2) Quantization: it refers to the process of compressing trained weight values in LLMs into lower bits (Nagel et al., 2020; Dettmers et al., 2022; Xiao et al., 2022; Frantar et al., 2022). Empirical evaluations have shown that int8 quantization can provide a great approximation of the predictive performance of the original LLMs (Dettmers et al., 2022). However, there is a significant decline in accuracy when attempting to reduce the number of bits even further.

## 2.3 PROMPT FOR LLMS

LLMs are known for their in-context learning ability, allowing them to generalize to unseen tasks without additional fine-tuning (Brown et al., 2020). Specifically, LLMs are controlled through user-provided natural language specifications of the task, or *prompts*, which illustrate how to complete a task. In this paradigm, we do not enforce modifications on the LLMs themselves. Instead, we focus on adapting the inputs to the LLMs for better predictive performance in downstream tasks. A typical strategy is to insert tokens before the input sequence to affect the attention mechanism. It has been shown in (Brown et al., 2020) that prompt engineering enables LLMs to match the performance of fine-tuned language models on a variety of language understanding tasks. Moreover, (Lester et al., 2021) empirically indicate that there is an equivalence between modifying the input and fine-tuning the model. Furthermore, (Su et al., 2022) studies the transferability of prompts across similar datasets or even tasks. Since then, we have witnessed the growth of prompt tuning infrastructure (Ding et al., 2022). However, we would like to emphasize that most of the current demonstrations of prompt tuning are task-specific (Li & Liang, 2021; Lester et al., 2021). When considering efficiency, it is desirable for a prompt to exhibit transferability across various settings.

## 3 MOTIVATION

The compression methods reduce the computational complexity at the cost of giving less accurate outputs. Thus, there naturally exists an **accuracy-efficiency trade-off**. In this section, we first empirically evaluate the trade-off of compressed LLMs. Then we found that for a compressed model, we can manually design a hard prompt that informs the model of its compressed state and helps it correct its predictions accordingly.

## 3.1 Performance of the Existing Approaches

**Experimental Setup.** We assess the trade-off using LLaMA (Touvron et al., 2023a) on C4 dataset (Raffel et al., 2020). Here we adopt two representative post-training compression methods, i.e., GPTQ (Frantar et al., 2022) and SparseGPT (Frantar & Alistarh, 2023), to analyze the trade-off across various compression levels. We note that we choose post-training compression methods primarily for their ease of deployment. For the quantization method, we apply GPTQ to compress the model weights into 2, 3, and 4 bits integer numbers. As for the pruning method, we employ SparseGPT to eliminate 50%, 62.5%, and 75% of the model parameters. We would like to note that the post-training compression is conducted using the training set of C4, and subsequently, we evaluate the performance of the compression with the validation set of C4.

**Quantitative Results.** As shown in Figure 2, we visualize the evaluation perplexity (PPL) (Jelinek et al., 1977) versus the compression level. When we prune 50% of the parameters or quantize the parameters to 4 bits, the PPL remains closer to that of the full LLaMA model. The PPL consistently increases as we decrease the allocated resource (e.g., bit-width/sparsity). Notably, the PPL will explode when the resource is below a certain threshold. For instance, the PPL shifts from 14 to 53 as sparsity increases from 62.5% to 75%. Moreover, the PPL grows significantly from around 11 to around 691 when we lower the quantization bits from 3-bit to 2-bit.

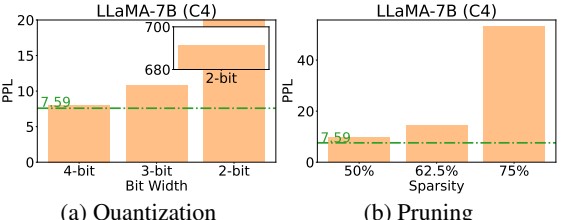

Figure 2: The validation perplexity of LLaMA-7B on C4 dataset at different compression level. The green line is the PPL of the original model.

**Qualitative Results.** As shown in the left part of Figure 1, besides PPL, we also do a case study to understand how compression affects model generation results. In this example, the full model is able to provide accurate and relevant answers to all three simple questions. Specifically, it correctly identifies Long Beach as a city in Los Angeles County, California, pinpoints Tulsa in northeastern Oklahoma, and describes asparagus as a spring vegetable belonging to the lily family. However, the pruned model with 62.5% weight sparsity struggles to generate meaningful responses. Instead of providing the requested information, its answers seem unrelated and tangential. For example, the pruned model responds with a statement about seeking a job when asked about Long Beach, mentions being a student at the University of Tulsa when asked about Tulsa's location, and admits uncertainty about Asparagus. This case study demonstrates that **aggressive model compression, such as the 62.5% weight sparsity applied to the pruned model, can lead to a significant degradation in the quality of generated responses.**

## 3.2 Prompt Compressed Models

In-context learning refers to the ability of adapting to the context provided within the input data through user-provided natural language specifications (Xie et al., 2022; Min et al., 2022), often referred to as *prompts*. Prompts serve to guide LLMs toward generating desired predictions by offering useful contextual information. As shown in Figure 1, the compressed model generates answers that are unrelated and off-topic when responding to these simple questions. Thus one natural question is, *for a compressed model, can we design a specific prompt that helps it correct its predictions accordingly?*

Following the question, we manually design the hard prompt as "*Please carefully examine the weight matrix within the model, as it may contain errors. It is crucial to verify its accuracy and make any necessary adjustments to ensure optimal performance*". The results are shown in the fourth column of Figure 1. The observations are summarized as follows:

**The prompted pruned model, i.e., "LLaMA-7B (62.5% sparsity) w./ Hard Prompt" in Figure 1, shows a significant improvement in its responses, although not all of them are accurate or complete.** Specifically, (1) when explicitly told about its compressed state, the prompted pruned model correctly identifies that Long Beach is located in the United States. However, it does not provide further information about the city, such as its presence in Los Angeles County, California. (2) Regarding the second question about Tulsa, Oklahoma, the prompted pruned model fails to provide a

relevant answer, instead repeating our prompt about the compression state, which is unrelated to the question. (3) When asked about asparagus, the prompted pruned model correctly identifies it as a plant used for cooking.

**Insights.** By explicitly informing the model of its compressed state, LLMs can generate more relevant responses for certain questions. The success of the designed prompt implies three great potentials:

1. **Cross-Dataset Transferability.** This human-designed prompt only provides the information that model weight is inaccurate. So intuitively, irrespective of the specific dataset being used, we hypothesize that the LLMs can generate more relevant responses with the same prompt.
2. **Cross-Compression Transferability.** Similarly, the human-designed prompt only mentions that the weight is inaccurate, without specifying the exact compression level or method. We hypothesize that LLMs can generate more relevant responses with the same prompt across different compression levels and methods.
3. **Cross-Task Transferability.** If LLMs can understand their compressed state and adjust accordingly, this adaptability is not limited to specific tasks or problem domains. Instead, it can be extended to a wide range of tasks.

However, despite the potential, as we analyzed at the beginning of this section, the manually designed prompt is not consistently effective. In other words, it only works for some problems, and not all answers generated are accurate or complete. Inspired by previous learnable prompt work (Li & Liang, 2021; Lester et al., 2021), we hypothesize that by involving the compressed weight in the prompt learning process, a learnable prompt could potentially surpass the performance of the hard prompt while still retaining the transferability aspects of the hard prompt.

## 4 LEARNING PROMPT FOR EFFICIENT LLM INFERENCE

In this section, we will begin by introducing the formulation of the prompt learning paradigm. Then, we will shift our focus to the maximum likelihood objective of learning the prompt. Finally, we will delve into the transferability of the learned prompts.

### 4.1 FORMULATION

Section 3.2 has shown that incorporating prompts can enhance the predictive performance of compressed LLMs. However, discovering effective language-based prompts through trial and error is a cumbersome and inefficient process that requires exploring a vast vocabulary space. Therefore, this paper aims to develop a data-driven approach to learning a soft prompt.

Typically an LLM would have a tokenizer that maps each input sentence into a sequence of integers $[x_0, x_1, \cdots, x_n]$. Afterwards, each token $x_i \in [v]$ represents a $d$-dimensional row vector in the embedding matrix $W \in \mathbb{R}^{v \times d}$. In the inference phase of LLM, we are given an input sequence $[x_0, x_1, \cdots, x_m]$ with $m$ tokens. We would like to generate tokens after $x_m$ step by step using an LLM. We denote prompt as a sequence of integers $[e_1, e_2, \cdots, e_k]$ with length $k$. Every token $e_j \in [k]$ represents a $d$-dimensional row vector in the prompt embedding matrix $E \in \mathbb{R}^{k \times d}$.

### 4.2 LEARNING OBJECTIVES

In this study, we present a prompt learning strategy that can be utilized as a post-training process for compressed LLMs. Given an LLM model with parameters denoted as $\theta$, we start with either sparsification (Frantar & Alistarh, 2023; Liu et al., 2023) or quantization (Frantar et al., 2022) approach that compresses the model parameters. We denote the parameters after the compression as $\widetilde{\theta}$. We note that prompt learning is reliant on the data, and as such, we need to employ a text dataset $X$ for this procedure. Next, for every sequence $[x_0, x_1, \cdots, x_n] \in X$, we insert $k$ prompt tokens $[e_1, e_2, \cdots, e_k]$ before it. Next, we optimize the following objective.

$$\min_E \mathcal{L}_{\widetilde{\theta}} = \min_E \sum_{t=1}^{n} -\log \Pr_{\widetilde{\theta}}[x_t | e_1, \cdots, e_k, x_0, \cdots x_{t-1}]. \tag{1}$$

We note that the model parameter $\widetilde{\theta}$ is fixed and not updated. And the trainable parameters are the embedding of the prompt tokens $[e_1, e_2, \cdots, e_k]$, which are denoted by the matrix $E \in \mathbb{R}^{k \times d}$.

Following (Lester et al., 2021), we initialize $E$ such that each row in $E$ corresponds to a vector randomly selected from the token embedding matrix $W$ of the LLM. The prompt token sequence remains the same for all sequences in $X$. This means that we use the representation of prompt tokens to influence LLM's attention mechanisms between the tokens in the sequence $[x_0, x_1, \cdots, x_n]$. Specifically, the Eq (1) aims to maximize the likelihood of correctly predicting the next token in the sequence, given the preceding tokens. In this way, the learned prompt is aware of the compressed weights, as the gradient flows through these compressed weights during the optimization process. This allows the model to adapt its behavior to account for the compression effects while generating responses, potentially leading to improved performance.

### 4.3 TRANSFERABILITY OF LEARNED PROMPT

The findings derived from Section 3.2 have provided us with a compelling impetus to delve into the exploration of the transferability of prompt tokens acquired through Eq (1). The representation of these prompt tokens, as well as their acquisition through one dataset, could have a significant impact on other NLP applications. Specifically, we have chosen to concentrate on the scenarios below.

**Cross-Dataset Transferability.** We aim to investigate whether prompt tokens trained from one dataset are applicable to other datasets. Prompt learning, while more efficient than fine-tuning, necessitates significant computational power and memory. With a single Nvidia-A100 possessing 40GB of memory, only the prompt learning of the LLaMA-7B model using a batch size of 1, sequence length of 1024, and 100 prompt tokens can be supported. If we perform a single round of prompt learning for a compressed LLM and achieve favorable outcomes across various datasets, we can substantially enhance the accuracy-efficiency trade-offs of the LLM during inference.

**Cross-Compression Transferability.** We aim to investigate the feasibility of utilizing learned prompts trained from a compressed LLM to another compressed LLM with different compression levels. For instance, we assess whether a prompt trained on a sparse LLM with a 75% sparsity can effectively boost the performance of an LLM with a 50% weight sparsity. Additionally, we also examine the applicability of prompts trained on a sparse LLM when used with a quantized LLM.

**Cross-Task Transferability.** We aim to investigate whether the learned prompt trained from Eq (1) on token generation tasks can be applied to other NLP tasks. This exploration will prove the effectiveness of prompts in improving the accuracy-efficiency trade-offs in the zero-shot generalization of LLMs in downstream tasks such as question answering.

## 5 EXPERIMENT

In this section, we assess the effectiveness of our prompt strategy in enhancing the trade-off between accuracy and efficiency during LLM inference. We commence by outlining the experimental setup, followed by presenting the results of token generation. Furthermore, we investigate the transferability of prompts across different datasets and compression levels. For additional experiments related to transferability and efficiency, please refer to Appendix A, where we have included further details.

### 5.1 EXPERIMENT SETTING

In our experimental framework, we incorporated the use of an Nvidia V100 GPU to conduct inference and prompt learning in LLMs. The datasets we utilized for token generation were comprehensive, including the Common Crawl's web corpus (C4) Raffel et al. (2020), Wikitext-2 Merity et al. (2017), and the Penn Treebank (PTB) Marcus et al. (1994) databases. We set the sequence length for these datasets to 1024. For the token generation task, we use perplexity (PPL) Jelinek et al. (1977) as the evaluation metric. We also introduce some downstream tasks to evaluate the cross-task transferability of the learned prompt. We will introduce the task information in the specific section. At the core of our modeling approach, we adopted the Open Pre-trained Transformer (OPT) Language Models (Zhang et al., 2022) and Large Language Model Architecture (LLaMA) (Touvron et al., 2023a). To compress the OPT and LLaMA model, we employed techniques from both SparseGPT (Frantar & Alistarh, 2023) and GPTQ (Frantar et al., 2022) methodologies. We refer the readers to Appendix A.1 for more experimental details.

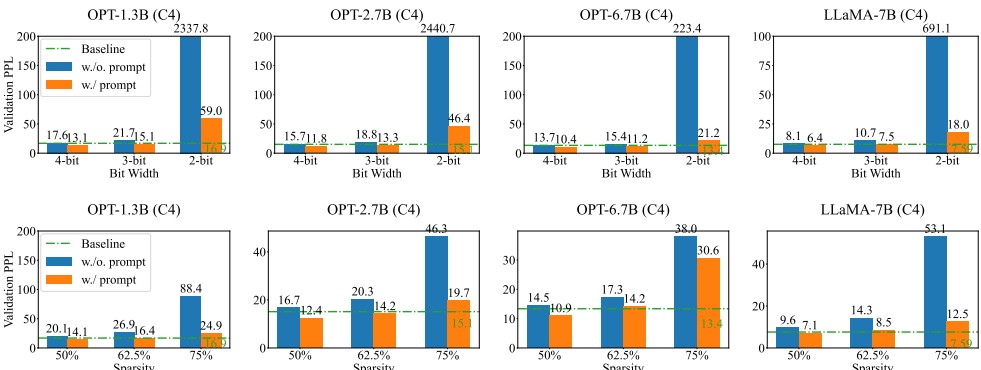

Figure 3: OPT-1.3B, OPT-2.7B, OPT-6.7B, and LLaMA-7B on C4 dataset, validation set at different bit-width and sparsity. Here the "Baseline" (green line) represents the uncompressed model.

## 5.2 TOKEN GENERATION RESULTS

On the C4 training set, we compress the OPT-1.3B, OPT-2.7B, OPT-6.7B, and LLaMA-7B using SparseGPT (Frantar & Alistarh, 2023). We utilize sparsity levels of 50%, 62.5%, and 75% for compression. Additionally, we employ GPTQ (Frantar et al., 2022) for 2-bit, 3-bit, and 4-bit quantization. Furthermore, prompt learning is applied to each compressed model using the methodology introduced in Eq (1). We set $k$ in Eq. 1 to 100, i.e., incorporating 100 learnable prompt tokens. We also conduct the ablation on the impact of the number of soft tokens in Appendix A.6. We note that the whole xprompt tuning process can be done in five hours with on four RTX 8000 48G GPUs.

Figure 3 demonstrates the impact of our approach on the validation set of C4. We observe a significant improvement in PPL across all compression levels. Firstly, by employing soft prompt tokens, the compressed LLMs using SparseGPT with 50% sparsity even outperform the full model counterparts, exhibiting lower PPL. This trend is also observed in the 4-bit quantization of LLMs using GPTQ. Secondly, even with further enhanced compression, the compressed LLMs with soft prompt tokens learned from Eq (1) still maintain comparable PPL to their original counterparts. Notably, prompts learned from each of the four 3-bit quantized models aid in surpassing the performance of their respective full model counterparts. We also observe a similar effect in sparse models with 62.5% sparsity for OPT-1.3B and OPT-2.7B. Conversely, prompts learned from both OPT-6.7B and LLaMA-7B assist in achieving the same PPL as their full model counterparts. Lastly, our approach significantly enhances the predictive performance of extreme scale compression. In both SparseGPT with 75% sparsity and GPTQ with 2-bit quantization, we find that the prompt learning strategy substantially improves the PPL across all four models. For example, prompts learned over the 2-bit GPTQ compression of OPT-1.3B reduce the PPL from 2337.8 to 59.

## 5.3 CROSS-DATASET TRANSFERABILITY

Intuitively, a model compressed using one dataset should achieve decent predictive performance when transferred to other datasets (Frantar et al., 2022; Frantar & Alistarh, 2023). Here we assess whether the prompt tokens learned from one dataset exhibit similar transferability across different datasets. Specifically, we first compress a model with SparseGPT or GPTQ using C4 training set. We then learn the prompt with the compressed model on C4 training set. Finally, we evaluate the performance of this compressed model with and without the learned prompts on other datasets, e.g., Wikitext-2 and PTB dataset. **We emphasize the entire process does not involve any task-specific data, and our results thus remain "zero-shot".**

Figure 4 presents the performance of OPT-1.3B, OPT-2.7B, OPT-6.7B, and LLaMA-7B on the test set of Wikitext-2 and the PTB dataset. For each LLM model, we also include the performance of its compressed versions with 50%, 62.5%, and 75% sparsity using SparseGPT. Additionally, we include the performance of each model's compressed version with 2-bit, 3-bit, and 4-bit quantization using GPTQ. The figures demonstrate the consistent advantages of prompt tokens across the two datasets. For every model with 50% sparsity or 4-bit quantization, learning prompts from the C4 dataset result in a lower PPL compared to the full model counterpart. Moreover, we observe a substantial

improvement in PPL when using learned prompt tokens as the model becomes more compressed. This phenomenon validates that the prompts learned on top of compressed models can be effectively transferred across datasets.

We also compare the transferred soft prompts against the soft prompts that are directly trained on the downstream dataset. Given direct prompt receives a domain-specific loss, our transferred prompt is, as expected, not as competitive as the direct one. However, such transferred prompt may significantly bridge the gap between a compressed and full model — e.g., our 3-bit & 4-bit quantized LLaMA-7B with transferred prompt can deliver on-par or better PPL than the full model on PTB and Wikitext2. We'd say this is an especially worthy contribution in practice, as one may possibly download the open-sourced transferable prompt to help on a compressed model with little effort.

Table 1: Perplexity comparison between full model and quantized models with different prompts, where we report test perplexity on PTB and Wikitext-2 dataset. "w./o. prompt" refers to the quantized model without soft prompts. "w./ direct prompt" means the soft prompts are directly trained on the target dataset. "w./ transferred prompt" means the prompt is trained on C4 dataset and then transferred to the target dataset.

| | Model | PTB | Wikitext2 |
|---|---|---|---|
| | Full Model | 11.02 | 6.33 |
| | Full Model w./ direct prompt | 6.86 | 5.57 |
| 4-bit | w./o. prompt | 11.65 | 6.92 |
| | w./ direct prompt | 7.04 | 5.88 |
| | w./ transferred prompt | 9.25 | 6.26 |
| 3-bit | w./o. prompt | 15.74 | 9.45 |
| | w./ direct prompt | 7.76 | 6.33 |
| | w./ transferred prompt | 10.81 | 6.90 |
| 2-bit | w./o. prompt | 5883.13 | 2692.81 |
| | w./ direct prompt | 14.98 | 16.67 |
| | w./ transferred prompt | 29.82 | 20.56 |

Here we emphasize that the prompt trained with a domain-specific loss can no longer be transferred between different datasets. Below we present the results of transferring the soft prompts learned on Wikitext2 ( featured articles on Wikipedia) to PTB (Wall Street Journal material) and C4 (collection of common web text corpus). The results, as shown in the table below, highlight a significant disparity in performance when using domain-specific prompts across different domains. The prompt trained on Wikitext-2, when applied to PTB and C4, leads to a drastic increase in perplexity, indicating a severe degradation in model performance. In contrast, if the prompt is learned on general text datasets like C4, then it can be transferred to different domains e.g., PTB and Wikitext2, and tasks, e.g., QA and language understanding (Appendix A.3).

Table 2: Perplexity comparison of transferring prompts learned on Wikitext2 to PTB and C4.

| Model | PTB | C4 |
|---|---|---|
| Full Model | 11.02 | 7.59 |
| 3-bit w./o. prompt | 15.74 | 10.74 |
| 3-bit w./ prompt learned on Wikitext2 | 294.16 | 160.64 |
| 3-bit w./ prompt learned on C4 | **10.81** | **7.48** |

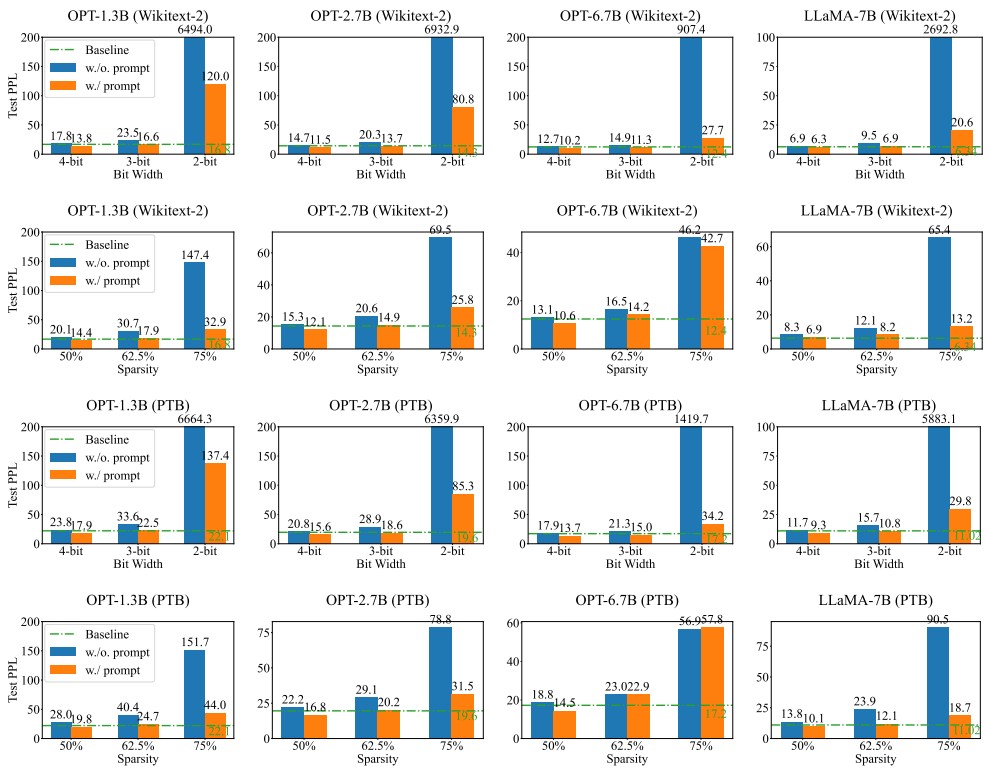

Figure 4: OPT-1.3B, OPT-2.7B, OPT-6.7B, and LLaMA-7B on Wikitext-2 and PTB test set at different bit-width and sparsity. Here the "Baseline" (green line) represents the uncompressed model.

## 5.4 COMBINATION OF SPARSIFICATION AND QUANTIZATION

In this section, we explore the effectiveness of the prompt strategy in the combination of sparsification and quantization for compressing LLM. Since sparsification and quantization target different aspects of compression, it is natural to combine them to achieve better efficiency. Table 3 presents the PPL before and with, and without the learned prompt on the validation set of C4, as well as the test sets of Wikitext-2 and PTB. We choose the LLaMA-7B model compressed using 50% sparsity and 4-bit quantization from the training set of C4. We should note that the prompt learning process also takes place on the training set of C4. Our results demon-

Table 3: The PPL of joint 50% sparsity + 4-bit quantization with learned prompts on the validation set of C4 and a test set of Wikitext-2 and PTB. The prompt is learned on C4 training set.

| Models | C4 | Wikitext-2 | PTB |
|--------|------|------------|-------|
| Full | 7.59 | 6.34 | 11.02 |
| 50% + 4-bit (w./o. prompt) | 10.94 | 9.67 | 17.39 |
| 50% + 4-bit (w./ prompt) | **7.38** | 7.31 | **10.64** |

strate that the prompt learning strategy remains effective when combining sparsification and quantization. Additionally, with the prompt, the 50% sparse and 4-bit compressed model still performs comparably to the original LLaMA-7B.

## 6 CONCLUSION

In this paper, we optimize the trade-off between computational efficiency and accuracy in LLMs via prompting compressed models. We propose a soft prompt learning method where we expose the compressed model to the prompt learning process. Our experimental analysis suggests our soft prompt strategy greatly improves the performance of the compressed models, allowing them to match their uncompressed counterparts. The research also highlights the transferability of these learned prompts across different datasets, tasks, and compression levels.

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

# Appendix

## A  MORE EXPERIMENTS

### A.1  EXPERIMENT DETAILS

In the experiment, we employed the AdamW Loshchilov & Hutter (2019) optimizer as our chosen optimizer. We conducted iterative prompt updates using a batch size of 4, a weight decay of $10^{-5}$, and a learning rate of $10^{-3}$. We set the total optimization steps as 30,000 and use the model corresponding to the best validation perplexity as the final model. To facilitate mix-precision training and system-level optimization, we leveraged the accelerate library Gugger et al. (2022).

All experiments are conducted on a server with eight Nvidia V100 (32GB) GPUs, 1.5T main memory, and two Intel Xeon CPU E5-2699A. The software and package version is specified below:

Table 4: Package configurations of our experiments.

| Package | Version |
|---|---|
| CUDA | 11.6 |
| pytorch | 2.0.1 |
| transformers | 4.30.0.dev0 |
| accelerate | 0.18.0 |

### A.2  ABLATION ON THE CROSS-COMPRESSION TRANSFERABILITY

Here we assess the transferability of learned prompts across various compression levels. Specifically, we aim to address the following questions: Can the prompt learned from a compressed model be applied to the same model but compressed at different levels or types?

In Figure 5, we display the Perplexity (PPL) outcomes on the C4 validation set, along with the results on the Wikitext-2 and PTB test sets. These results are obtained by applying prompts learned from a source compressed model to a different target compressed model. Here, "target" denotes the specific compression type and degree used in the model receiving the prompts. While "source" refers to the compression type and degree of the model from which the prompts are originally learned. For example, "source 4-bit" indicates that the prompt is learned from a compressed model with 4-bit quantization. Based on the figures, we observe that (1) For sparse LLMs, prompts learned from higher sparsity can be effectively transferred to models with lower sparsity, while achieving comparable performance.. (2) For quantized LLMs, prompts learned from lower bit quantization levels can be successfully applied to models with higher bit quantization, while achieving comparable performance. (3) There is a certain degree of transferability of prompts learned between different compression types, especially when the compression level is less. For instance, a prompt learned from a LLaMA-7B model with 4-bit quantization can be transferred to a LLaMA-7B model with 50% sparsity.

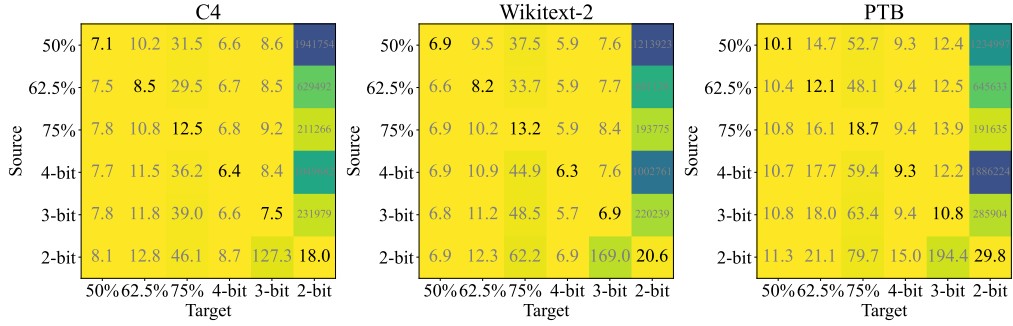

Figure 5: LLaMA-7B transfer between different sparsity and bit-width. The "target" refers to the compression type and level for the compressed model, while the "source" represents the type and level of the compressed model from which the prompt is learned. For example, "4-bit" in source indicates that the prompt is learned from a compressed model with 4-bit quantization.

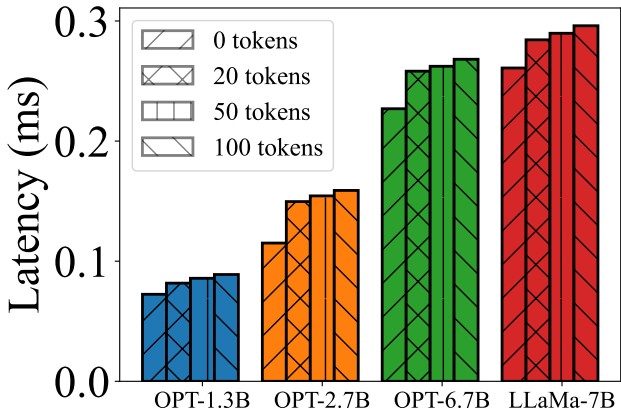

Figure 6: Latency benchmark of inference speed with prompt tokens

### A.3 CROSS-TASK TRANSFERABILITY

In this section, we explore the transferability of learned prompts across different tasks. Specifically, we aim to assess the effectiveness of prompts learned from token generation tasks, as indicated by Eq (1), in downstream tasks of LLM. As an illustrative example, we consider the zero-shot generalization tasks of LLaMA-7B Touvron et al. (2023a). For evaluation purposes, we have chosen OpenbookQA Mihaylov et al. (2018), Hellaswag Zellers et al. (2019), PIQA Bisk et al. (2020), and the high school European history task from Hendrycks et al. (2020). The European history task is particularly interesting due to its inclusion of a lengthy context sentence for each question. We employ the lm-evaluation-hardness framework Gao et al. (2021), incorporating adapters from Yuan et al. (2022), for the purpose of conducting the experiment.

Table 5 presents the results in terms of normalized accuracy, and we also include the standard deviation, as indicated by Gao et al. (2021). The table clearly demonstrates that the learned prompt significantly enhances the accuracy of these tasks. These findings imply that prompts acquired through token generation tasks can effectively enhance the accuracy-efficiency trade-off of compressed LLMs.

### A.4 EFFICIENCY PROFILING

In this section, we analyze how the inclusion of prompt tokens impacts the latency of LLM inference. Figure 6 illustrates the latency of three OPT models and the LLaMA-7B model utilized in this paper, considering the insertion of additional prompt tokens with varying lengths. For token generation, we set the sequence length to 1024. The figure demonstrates that the addition of prompt tokens does not significantly increase the latency of LLM inference, particularly when the inserted tokens account for less than 10% of the original sequence length. Furthermore, our observations indicate that the latency does not exhibit a linear correlation with the length of the inserted tokens, highlighting the effectiveness of the prompt in facilitating efficient LLM inference.

### A.5 MORE EXPERIMENTS ON LARGE LLMS

We perform our methods and LoRA (Hu et al., 2021) on LLaMA-2-13B and bloom-7B models. The results are summarized on Table 6. Following LoRA experimental setting (Hu et al., 2021), we insert LoRA layers to all query and value layers with a rank $r = 32$, $\alpha = 32$, and a $0.1$ dropout rate. We train the Lora using the Adam optimizer with a $2e - 4$ learning rate. It suggests that our approach outperforms LoRA with lower PPL. Moreover, we can recover the performance of GPTQ 3-bit LLaMA-2-13B and GPTQ 3-bit BLOOM-7B with even better performance than their fp16 counterparts.

We also test the transferrability of ours and LoRA on wikitext2 dataset. We summarize the results in Table 7. It suggests that our soft prompt can be transferred to other dataset while still outperforming LoRA for the performance recovery of compressed LLMs.

Table 5: The zero-shot results on transforming the learned prompt to OpenBookQA, Hellaswag, PIQA, and High School European History dataset.

| Models | | OpenbookQA | Hellaswag | PIQA | High School European History |
|---|---|---|---|---|---|
| Full | | 0.410±0.022 | 0.497±0.005 | 0.702±0.011 | 0.364±0.038 |
| 50% | w./o. Prompt | 0.412±0.022 | 0.449±0.005 | 0.682±0.011 | 0.364±0.038 |
| | + Learned Prompt | 0.400±0.022 | 0.469±0.005 | 0.689±0.011 | 0.358±0.037 |
| 62.5% | w./o. Prompt | 0.396±0.022 | 0.380±0.005 | 0.638±0.011 | 0.345±0.037 |
| | + Learned Prompt | 0.402±0.022 | 0.433±0.005 | 0.668±0.011 | 0.345±0.037 |
| 75% | w./o. Prompt | 0.366±0.022 | 0.280±0.004 | 0.549±0.012 | 0.315±0.036 |
| | + Learned Prompt | 0.358±0.021 | 0.344±0.005 | 0.614±0.011 | 0.358±0.037 |
| 4-bit | w./o. Prompt | 0.410±0.022 | 0.487±0.005 | 0.690±0.011 | 0.358±0.037 |
| | + Learned Prompt | 0.418±0.022 | 0.487±0.005 | 0.692±0.011 | 0.352±0.037 |
| 3-bit | w./o. Prompt | 0.378±0.022 | 0.446±0.005 | 0.674±0.011 | 0.358±0.037 |
| | + Learned Prompt | 0.404±0.022 | 0.459±0.005 | 0.688±0.011 | 0.358±0.037 |
| 2-bit | w./o. Prompt | 0.354±0.021 | 0.240±0.004 | 0.491±0.012 | 0.315±0.036 |
| | + Learned Prompt | 0.350±0.021 | 0.294±0.005 | 0.563±0.012 | 0.333±0.037 |

Table 6: The validation PPL of Llama-2-13B and Bloom-7B models on C4 dataset.

| Dataset | Model | Precision | Recover method | Trainable Params (M) | PPL |
|---|---|---|---|---|---|
| C4 | Llama-2-13B | fp16 | NA | NA | 6.96 |
| C4 | Llama-2-13B | 3bit | NA | NA | 9.24 |
| C4 | Llama-2-13B | 3bit | Soft Prompt | 0.5 | **6.75** |
| C4 | Llama-2-13B | 3bit | LoRA | 26 | 8.15 |
| C4 | bloom-7B | fp16 | NA | NA | 15.87 |
| C4 | bloom-7B | 3bit | NA | NA | 18.40 |
| C4 | bloom-7B | 3bit | Soft Prompt | 0.4 | **13.54** |
| C4 | bloom-7B | 3bit | LoRA | 15.7 | 17.26 |

### A.6 ABLATION ON THE NUMBER OF SOFT TOKENS

In Table 8, we conduct the ablation study on the impact of the number of soft tokens using 3-bit quantized LLama-7B on PTB dataset. We observe that there is still a significant improvement with 25 prompt tokens, and we can improve the performance by increasing the prompt size.

## B DISCUSSION

**Limitations.** One limitation of our study is its reliance on GPUs for executing computational tasks. It is crucial to acknowledge that GPUs can be expensive to procure and maintain, thus imposing financial constraints on researchers or organizations with limited resources. In order to address this issue, future endeavors should investigate alternative computational architectures or optimizations that can alleviate the dependence on costly GPUs. By doing so, the accessibility and applicability of our proposed methodology can be expanded, making it more widely accessible to a broader range of researchers and organizations.

**Potential Negative Societal Impacts.** While our research primarily centers on diminishing the energy consumption of LLM during inference, it is crucial to acknowledge that the carbon emissions stemming from LLM inference may still contribute to environmental concerns. As part of our future endeavors, we aspire to enhance the efficiency of LLM inference on low-energy devices.

## C MORE VISUALIZATION

In this section, we present further visualizations of compression-aware prompts, as demonstrated in Figure 1 in Section 1. The results unveil a significant improvement achieved by utilizing a hard,

Table 7: The zero-shot test PPL of transferred soft prompt and Lora on Wikitext2 dataset.

| Dataset | Model | Precision | Method | Transferred Params (M) | PPL |
|---|---|---|---|---|---|
| Wikitext2 | Llama-2-13B | fp16 | NA | NA | 5.58 |
| Wikitext2 | Llama-2-13B | 3bit | NA | NA | 7.88 |
| Wikitext2 | Llama-2-13B | 3bit | Soft Prompt | 0.5 | **5.89** |
| Wikitext2 | Llama-2-13B | 3bit | LoRA | 26 | 7.07 |
| Wikitext2 | BLOOM-7B | fp16 | NA | NA | 13.26 |
| Wikitext2 | BLOOM-7B | 3bit | NA | NA | 16.06 |
| Wikitext2 | BLOOM-7B | 3bit | Soft Prompt | 0.4 | **12.42** |
| Wikitext2 | BLOOM-7B | 3bit | LoRA | 15.7 | 15.65 |

Table 8: Ablation study on the impact of the number of soft tokens using 3-bit quantized LLama-7B on PTB dataset.

| # tokens | Perplexity |
|---|---|
| Baseline (0 tokens) | 15.74 |
| 25 tokens | 9.26 |
| 50 tokens | 8.61 |
| 75 tokens | 8.17 |
| 100 tokens | 7.76 |

task-independent prompt on compressed LLMs. Additionally, we showcase the visualization of responses generated using our prompt derived from the C4 training set. It is worth noting that, in certain instances, the task-independent and learned prompt outperforms the hard prompt.

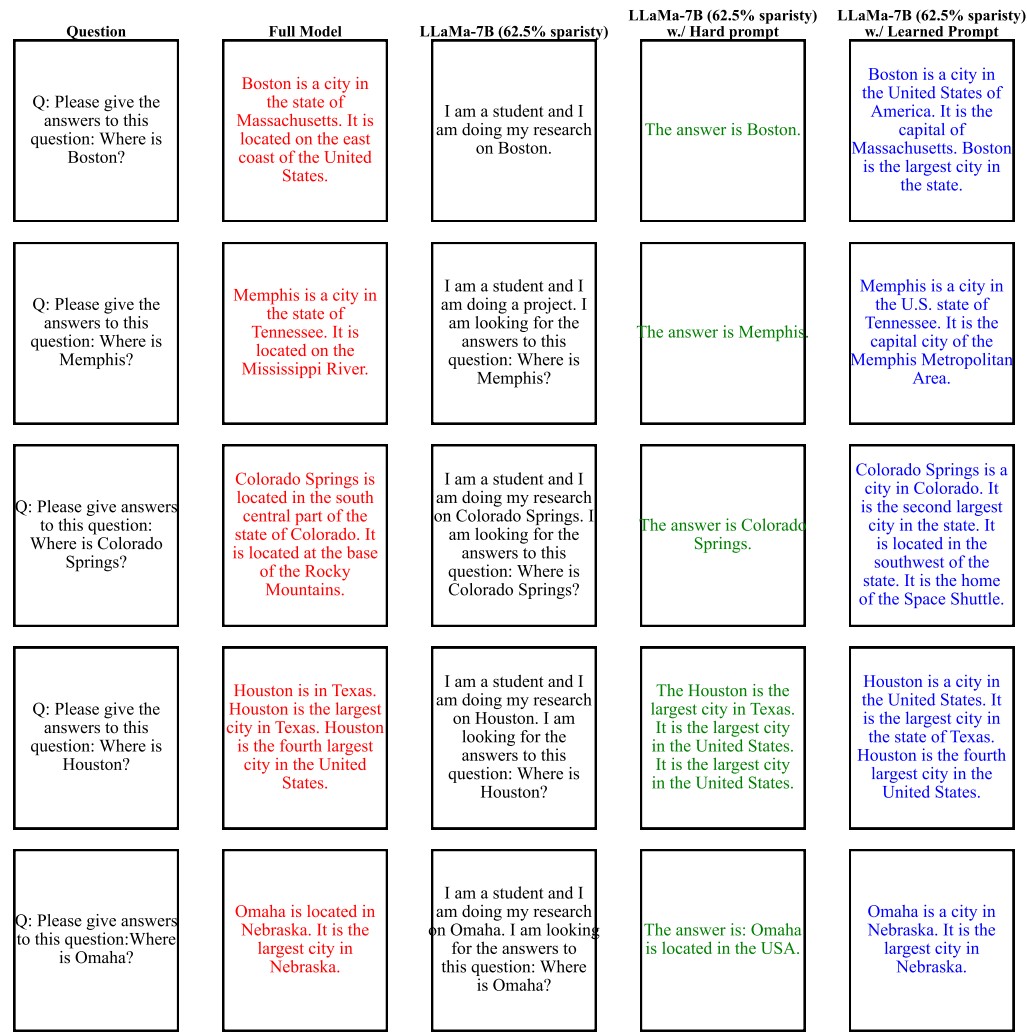

Figure 7: Case study for the effect of prompts on a pruned LLaMA-7B with a 62.5% weight sparsity.

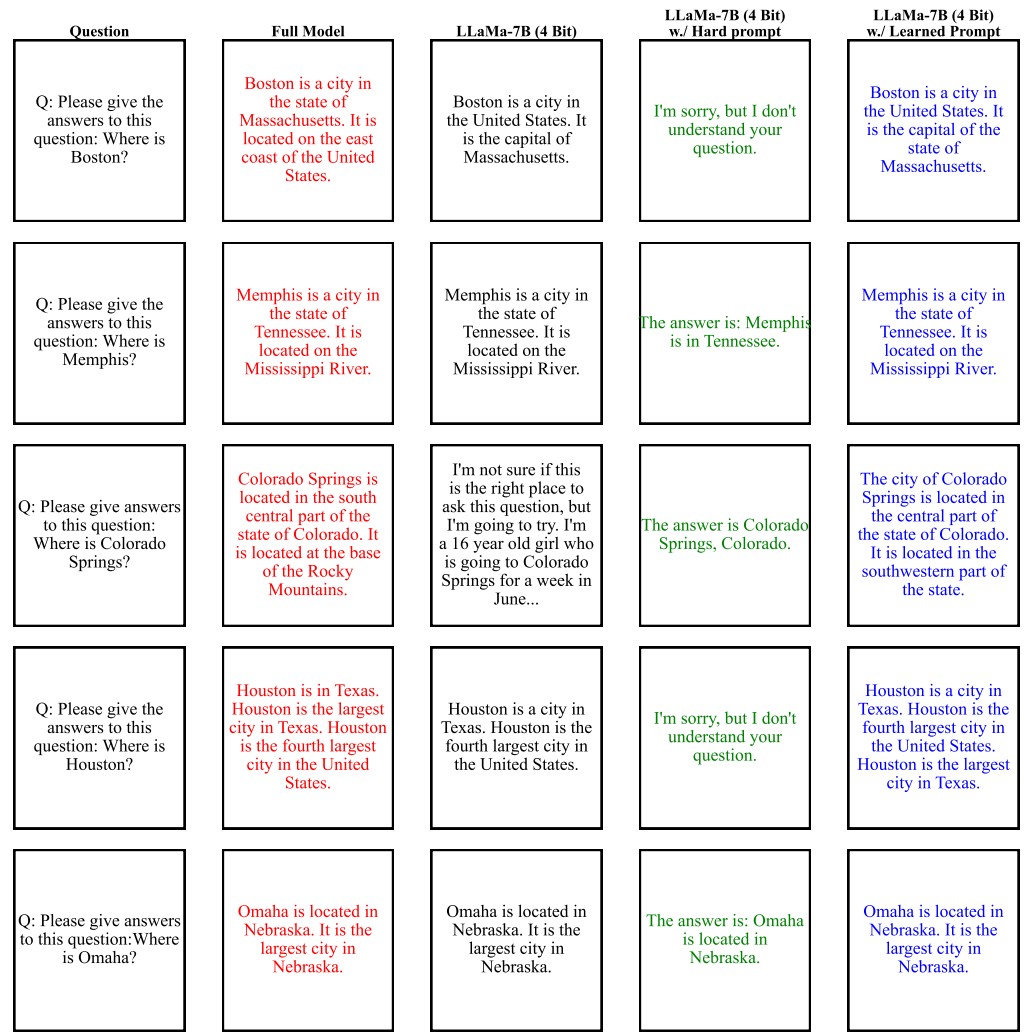

Figure 8: Case study for the effect of prompts on a pruned LLaMA-7B with a 4-bit quantization.

# D UNDERSTANDING THE LEARNED PROMPTS FROM NATURAL LANGUAGE ASPECT

With the learned prompt outperforming the hard counterpart, we raise an intriguing question: How do the learned prompt tokens look when viewed from the perspective of natural language? In this section, we present the ablation study to answer the above question. Specifically, for each of the learned prompt token embeddings, we identify the words whose embedding is closest to the learned prompt token embedding via the nearest neighbor search technique, where the similarity measure is cosine similarity. In Figure 9, we plot the histogram of the cosine similarity between each learned prompt token and the top-100 nearest embeddings to it, where the prompt is learned with a pruned LLaMA-7B with a 50% weight sparsity. **We observe that there is no word whose embedding closely matches the learned one within the embedding space.** The cosine similarity for nearly all comparisons falls below 0.16, suggesting a considerable disparity between the learned prompt embeddings and their nearest equivalents. Below we also report the nearest word for each of the learned prompt token embedding. We observe that (1) almost all of them are meaningless. (2) several learned prompt tokens may be mapped to the same word.

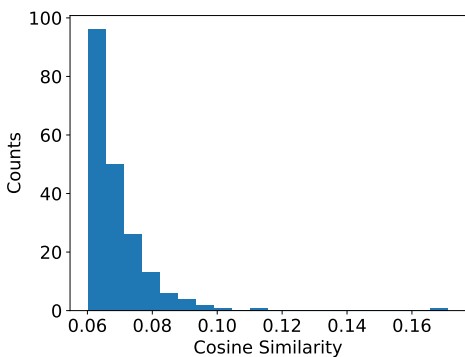

Figure 9: The distribution of the cosine similarity between the learned embedding and the top-100 nearest embeddings to it.

**Nearest word for each of the learned prompt token[i]:** `"heits"`, `""`, `""`, `"<0x00>"`, `"<0x01>"`, `"<0x02>"`, `"<0x03>"`, `"<0x04>"`, `"<0x05>"`, `"<0x06>"`, `"<0x07>"`, `"<0x08>"`, `"<0x09>"`, `"<0x0A>"`, `"<0x0B>"`, `"<0x0C>"`, `"<0x1A>"`, `"<0x0E>"`, `"<0x0F>"`, `"<0x10>"`, `"<0x11>"`, `"<0x12>"`, `"<0x13>"`, `"<0x14>"`, `"<0x15>"`, `"<0x16>"`, `"<0x17>"`, `"<0x18>"`, `"<0x19>"`, `"<0x1A>"`, `"<0x1B>"`, `"<0x1C>"`, `"<0x1D>"`, `"<0x1E>"`, `"<0x1F>"`, `"sep"`, `";;;;"`, `"état"`, `"<0xB1>"`, `"_Ej"`, `"moz"`, `"_diverse"`, `"_"`, `"argument"`, `"|"`, `"han"`, `"ura"`, `"/"`, `"-"`, `"<0xE7>"`, `"_Lisa"`, `"_case"`, `"ura"`, `"O"`,`"_Chal"`, `"_Chan"`, `"O"`, `"asc"`, `"Client"`, `"_Det"`, `"O"`, `"_Hel"`, `"_L"`, `"_Pel"`, `"_k"`, `"_It"`, `"O"`, `"<0x8B>"`, `"<0x00>"`, `"ILL"`, `"O"`, `"E"`, `"ren"`, `"ety"`, `"cy"`, `""`, `"<0x8B>"`, `"<0x9F>"`, `""`, `""`, `"IM"`, `""`, `"."`

*Our ablation study highlights the hardness of understanding the mechanisms underlying learned prompts. This area remains largely uncharted, inviting future research to uncover its intricacies.* Our hope is that this study will ignite curiosity and foster continued scholarly pursuit in this field.

---

[i]here we did not display the word that is not in UTF-8 format.

