# OpenReview forum: "Compress, Then Prompt: Improving Accuracy-Efficiency Trade-off of LLM Inference with Transferable Prompt"
_ICLR.cc/2024/Conference — Submitted to ICLR 2024_

### Official Review · Reviewer_jG4W · 2023-10-29

**Soundness:** 3 good
**Presentation:** 3 good
**Contribution:** 2 fair
**Rating:** 5
**Confidence:** 4

**Summary:**

In this paper, the authors first identify the efficiency problems with the inference process of Large Language Models (LLMs). Then, the authors find that quantization and pruning are utilized to deal with the efficiency problem while incurring performance issues, i.e., the PPL of the model is high. Then, the authors propose a soft prompt approach to improve the performance of the quantized or pruned model. In addition, the authors validate three aspects of the proposed methods, i.e., cross dataset transferability, cross compression transferability, and cross-task transferability.

**Strengths:**

1. The approach of soft prompt to address the performance of the quantized or pruned LLM is effective.
2. The analysis of the three aspects of the proposed approach is interesting.
3. The paper is well organized and easy to follow.

**Weaknesses:**

1. The approach is simple and the novelty is not obvious. The soft prompt method is already proposed in multiple existing papers.
2. The evaluation is only conducted with PPL. However, the evaluation of LLM should be well designed and other methods should be exploited to further validate the proposed method.
3. More tasks should be examined to show the effectiveness of the proposed approach.

**Questions:**

1. I wonder if there are other LLM evaluation methods besides PPL.
2. I wonder it the method can be applied to other tasks, e.g., captioning.
3. I wonder what would be the difference between the proposed methods and conventional soft prompt tuning except using different LLMs, i.e., one with quantized or pruned model while the other one with the original model.

---

> ### Author Response · Authors · 2023-11-19
> **Thanks & Initial Response to Reviewer jG4W (1/2)**
>
> ### **[W1 - The approach is simple and the novelty is not obvious: Our paper uncovers new properties of prompt tuning and they offer new opportunities for model compression.]**
>
> While novelty is a multifaceted concept in academic research, we believe it can be roughly viewed from two fronts: ***empirical novelty***, which involves uncovering new properties and behaviors unknown to the wider community (e.g., the emergent ability [1], lottery ticket hypothesis [2]), and ***technical novelty***, which pertains to the development of new solutions or methodologies (e.g., LoRA [3]).
>
> We argue our work is rich in empirical novelty in at least two aspects:
>
> 1. **Transformation of Prompt Tuning from Task-Specific to Transferable:** Prior to our study, only a few works studied the transferability of learned prompts between different tasks [4, 5, 6]. Specifically, [4] finds it is possible to transfer learnable prompts with **additional fine-tuning on downstream task**. However, as mentioned in the experiment section, all our reported results are **zero-shot**, i.e., no fine-tuning is needed to obtain transferability. Furthermore, [5] finds that the learned prompt can only be transferred among similar tasks. However, as verified by our experiments, our learned prompts are transferable between datasets, tasks, and compressed models.
>
> 2. **New avenue to enhance Compressed Model Accuracy:** We show that prompt tuning can effectively recover the accuracy drop of compressed models. Traditional model compression methods often demand extensive engineering efforts and intricate designs, as seen in popular model compression papers like GPTQ [7], SPQR [8], and AWQ [8]. However, our method simplifies this process significantly. We leverage prompt tuning to effectively recover the accuracy drop in compressed models, opening a new avenue to optimize the trade-off between accuracy and efficiency.
>
> These findings are not only valid but also generalizable across various model families, datasets, and tasks, underscoring the broad applicability and impact of our work.
>
> Regarding technical contribution, we agree that our method is a direct application of prompt tuning. However, we emphasize that **we intensionally kept our method as simple as possible**. This simplicity, is in our opinon more of a strength than a weakness, as it just underscores the unknown properties of straightforward prompt tuning methods; where the uncovering of these properties and the new pathways they opened for model compression constitute the true novelty of our paper. Thereby, we argue that we contribute substantially to the field's understanding of prompt tuning and model compression.
>
>
>
>
> [1] Emergent Abilities of Large Language Models
>
> [2] The Lottery Ticket Hypothesis: Finding Sparse, Trainable Neural Networks
>
> [3] LoRA: Low-Rank Adaptation of Large Language Models
>
> [4] On Transferability of Prompt Tuning for Natural Language Processing
>
> [5] SPoT: Better Frozen Model Adaptation through Soft Prompt Transfer
>
> [6] Reducing Retraining by Recycling Parameter-Efficient Prompts
>
> [7] GPTQ: Accurate Post-Training Quantization for Generative Pre-trained Transformers
>
> [8] SpQR: A Sparse-Quantized Representation for Near-Lossless LLM Weight Compression
>
> [9] AWQ: Activation-aware Weight Quantization for LLM Compression and Acceleration

---

> ### Author Response · Authors · 2023-11-19
> **Thanks & Initial Response to Reviewer jG4W (2/2)**
>
> ### **[W2,Q1,Q2 - The evaluation is only conducted with PPL: We have in fact evaluated on other tasks]**
>
> **We kindly direct the reviewer's attention to [Appendix A.3](https://openreview.net/pdf?id=Gdm87rRjep#page=15)** , where we have explored QA (OpenBookQA, PIQA), commonsense NLI (Hellaswag), language understanding (high school European history from MMLU) tasks reported with non-PPL metrics. We highlight that all these reported results are zero-shot. Namely, the prompt is learned on C4, and then we directly stitch the prompt to the compressed model and test the performance on these downstream tasks. We hope the reviewer may find them helpful.
>
>
> | Models |                  | OpenbookQA      | Hellaswag       | PIQA            | High School European History |
> |--------|------------------|-----------------|-----------------|-----------------|------------------------------|
> | Full   |                  | 0.410±0.022     | 0.497±0.005     | 0.702±0.011     | 0.364±0.038                  |
> | 50%    | w./o. Prompt     | 0.412±0.022     | 0.449±0.005     | 0.682±0.011     | **0.364±0.038**              |
> |        | + Learned Prompt | 0.400±0.022     | **0.469±0.005** | **0.689±0.011** | 0.358±0.037                  |
> | 62.5%  | w./o. Prompt     | 0.396±0.022     | 0.380±0.005     | 0.638±0.011     | 0.345±0.037                  |
> |        | + Learned Prompt | **0.402±0.022** | **0.433±0.005** | **0.668±0.011** | 0.345±0.037                  |
> | 75%    | w./o. Prompt     | 0.366±0.022     | 0.280±0.004     | 0.549±0.012     | 0.315±0.036                  |
> |        | + Learned Prompt | 0.358±0.021     | **0.344±0.005** | **0.614±0.011** | **0.358±0.037**              |
> | 4-bit  | w./o. Prompt     | 0.410±0.022     | 0.487±0.005     | 0.690±0.011     | 0.358±0.037                  |
> |        | + Learned Prompt | **0.418±0.022** | 0.487±0.005     | **0.692±0.011** | 0.352±0.037                  |
> | 3-bit  | w./o. Prompt     | 0.378±0.022     | 0.446±0.005     | 0.674±0.011     | 0.358±0.037                  |
> |        | + Learned Prompt | **0.404±0.022** | **0.459±0.005** | **0.688±0.011** | 0.358±0.037                  |
> | 2-bit  | w./o. Prompt     | 0.354±0.021     | 0.240±0.004     | 0.491±0.012     | 0.315±0.036                  |
> |        | + Learned Prompt | 0.350±0.021     | **0.294±0.005** | **0.563±0.012** | **0.333±0.037**              |
>
>
>
> ### **[W3 - The difference between the proposed methods and conventional soft prompt tuning: Transformation of Prompt Tuning from Task-Specific to Transferable]**
>
>
> As we mentioned in our response to W1 above, the key difference between our prompt tuning and conventional soft prompt tuning is that our learned soft prompts are transferable between datasets, tasks, and compression levels. The transferring of prompts saves the prompt tuning time and effort originally required for each specific setups (according to our experience with 4 GPUs, that means 5-ish hours per task per compressed model).
>
> We also already compared our methods with conventional soft prompt tuning in [experiment](https://openreview.net/pdf?id=Gdm87rRjep#page=8) (**NOTE**: it was originally in Table 4 of Appendix A.3, we move it to the main text in the updated version). For your convenience, we post our results and summarize our observations here:
>
>
> > Perplexity comparison between full model and quantized models with different prompts. Where we report test perplexity on PTB and Wikitext-2 dataset. "w./o. prompt" refers to the quantized model without soft prompts. "w./ direct prompt" means the soft prompts are directly trained on the target dataset. "w./ transferred prompt" means the prompt is trained on C4 dataset and then transferred to the target dataset.
>
> | Model                       | PTB     | Wikitext2 |
> |-----------------------------|---------|-----------|
> | Full Model                  | 11.02   | 6.33      |
> | Full Model w./ direct prompt| 6.86    | 5.57      |
> | 4-bit w./o. prompt          | 11.65   | 6.92      |
> | 4-bit w./ direct prompt     | 7.04    | 5.88      |
> | 4-bit w./ transferred prompt| 9.25    | 6.26      |
> | 3-bit w./o. prompt          | 15.74   | 9.45      |
> | 3-bit w./ direct prompt     | 7.76    | 6.33      |
> | 3-bit w./ transferred prompt| 10.81   | 6.90      |
> | 2-bit w./o. prompt          | 5883.13 | 2692.81   |
> | 2-bit w./ direct prompt     | 14.98   | 16.67     |
> | 2-bit w./ transferred prompt| 29.82   | 20.56     |
>
> Given direct prompt receives a task-specific loss, our transferred prompt is, as expected, not as competitive as the direct one. However, such transferred prompt may significantly bridge the gap between a compressed and full model — e.g., our 3-bit & 4-bit quantized LLaMA-7B with transferred prompt can deliver on-par or better PPL than the full model on PTB and Wikitext2. We'd say this is an especially worthy contribution in practice, as one may possibly download the open-sourced transferable prompt to help on a compressed model with little effort.

---

### Official Review · Reviewer_34kz · 2023-10-30

**Soundness:** 3 good
**Presentation:** 2 fair
**Contribution:** 2 fair
**Rating:** 5
**Confidence:** 4

**Summary:**

In order to restore the performance of compressed models (either quantized or weight sparsified, or both), this paper applies prefix tuning on compressed models, and studies whether a prefix decided on one dataset, one compression rate, or one task can generalize to others. This paper experiments on compressed OPT (1.3B, 2.7B, 6.7B) and LLaMA (7B) models, using C4, Wikitext-2, and Penn Treebank for perplexity evaluation, and OpenbookQA, Hellaswag, PIQA, and HSEH for zero-shot accuracy evaluation.

**Strengths:**

* The research question of how to make LLMs smaller and maintain their performance is very important.
* Tuning the prompt prefix is a reasonable way to help with that.
* The paper has empirically studied the transferability of the learned prefix to other compression rates and datasets.

**Weaknesses:**

* Novelty is limited. The literature has applied the soft prefix tuning method to uncompressed models, and this paper applies the soft prefix tuning method to a compressed model. It's like a verification scenario of the prefix tuning method.
* Experimental verification needs to be improved:
  - Experiments are only conducted on small and relatively old models (4 OPT and LLaMA-v1, all <7B).
  - Do not compare with other strategies of finetuning in model compression, e.g., how about we apply LoRA tuning to restore the performance of model compression?

## Minor
* There are some minor writing issues, need some proofreading:
  - Unify “PPL” and “Perplexity” in Figure 2.
  - No caption for Figure 6.
  - One result is wrongly colored in Table 5. 50% row, HSEH column.

**Questions:**

* Can the prompt prefix generalize across different models?

---

> ### Author Response · Authors · 2023-11-19
> **Thanks & Initial Response to Reviewer 34kz (1/2)**
>
> We thank the reviewer for the comments and suggestions to improve our paper. Please see the following clarifications.
>
> ### **[W1 - Novelty is limited: Our paper uncovers new properties of prompt tuning and they offer new opportunities for model compression]**
>
>
> While novelty is a multifaceted concept in academic research, we believe it can be roughly viewed from two fronts: ***empirical novelty***, which involves uncovering new properties and behaviors unknown to the wider community (e.g., the emergent ability [1], lottery ticket hypothesis [2]), and ***technical novelty***, which pertains to the development of new solutions or methodologies (e.g., LoRA [3]).
>
> We argue our work is rich in empirical novelty in at least two aspects:
>
> 1. **Transformation of Prompt Tuning from Task-Specific to Transferable:** Prior to our study, only a few works studied the transferability of learned prompts between different tasks [4, 5, 6]. Specifically, [4] finds it is possible to transfer learnable prompts with **additional fine-tuning on downstream task**. However, as mentioned in the experiment section, all our reported results are **zero-shot**, i.e., no fine-tuning is needed to obtain transferability. Furthermore, [5] finds that the learned prompt can only be transferred among similar tasks. However, as verified by our experiments, our learned prompts are transferable between datasets, tasks, and compressed models.
>
> 2. **New avenue to enhance Compressed Model Accuracy:** We show that prompt tuning can effectively recover the accuracy drop of compressed models. Traditional model compression methods often demand extensive engineering efforts and intricate designs, as seen in popular model compression papers like GPTQ [7], SPQR [8], and AWQ [8]. However, our method simplifies this process significantly. We leverage prompt tuning to effectively recover the accuracy drop in compressed models, opening a new avenue to optimize the trade-off between accuracy and efficiency.
>
> These findings are not only valid but also generalizable across various model families, datasets, and tasks, underscoring the broad applicability and impact of our work.
>
> Regarding technical contribution, we agree that our method is a direct application of prompt tuning. However, we emphasize that **we intensionally kept our method as simple as possible**. This simplicity, is in our opinon more of a strength than a weakness, as it just underscores the unknown properties of straightforward prompt tuning methods; where the uncovering of these properties and the new pathways they opened for model compression constitute the true novelty of our paper. Thereby, we argue that we contribute substantially to the field's understanding of prompt tuning and model compression.
>
>
>
>
> [1] Emergent Abilities of Large Language Models
>
> [2] The Lottery Ticket Hypothesis: Finding Sparse, Trainable Neural Networks
>
> [3] LoRA: Low-Rank Adaptation of Large Language Models
>
> [4] On Transferability of Prompt Tuning for Natural Language Processing
>
> [5] SPoT: Better Frozen Model Adaptation through Soft Prompt Transfer
>
> [6] Reducing Retraining by Recycling Parameter-Efficient Prompts
>
> [7] GPTQ: Accurate Post-Training Quantization for Generative Pre-trained Transformers
>
> [8] SpQR: A Sparse-Quantized Representation for Near-Lossless LLM Weight Compression
>
> [9] AWQ: Activation-aware Weight Quantization for LLM Compression and Acceleration

---

> > ### Comment · Reviewer_34kz · 2023-11-22
> > **Thanks for the response & Follow-up comments**
> >
> > I thank the authors for the detailed response and the added experiment! I'm still concerned about the technical or empirical novelty.
> >
> > First, I hold a different opinion regarding the "New avenue to enhance Compressed Model Accuracy" point. technically, this work is a prompt fine-tuning method, which should be put under the same context as LoRA-based tuning, rather than saying it's simpler than orthogonal work like GPTQ or AWQ.
> >
> > Second, if the authors would like to mainly highlight the empirical evaluation of this paper. Then, my follow-up suggestions are:
> > 1. Need more explorations and analyses to make the empirical observations solid and actually reveal useful knowledge. Just to name a few, can we answer the following questions:
> >    * What is the LoRA hyperparameter used in the newly added experiment? Did we sweep the hyperparameters of both the prompt tuning and LoRA method? This can make the current results valid.
> >    * Is prompt tuning better than LoRA for uncompressed models? If not, why is prompt tuning so much better than LoRA for compressed models? If it is due to the parameter size, can we adjust the hyperparameter or tensor choice of LoRA to reduce the parameter size?
> >    * For restoring the performance of compressed models, is prompt tuning universally better for all types of tasks? Or prompt tuning is only better on several types of compressed models?
> >    * Did the authors want to claim that in the future, we don't need LoRA when we are compressing models? No matter what task we are targeting, we should use prompt tuning as it achieves much better performance?
> > 2. It's better for this paper to be re-organized to correctly position the empirical evaluation contribution: First, summarize valuable questions that engineers might care about the performance restoration of compressed models; Then, conduct extensive results to reveal phenomena, and analyze why; Finally, give answers to the important questions an suggestions.

---

> ### Author Response · Authors · 2023-11-19
> **Thanks & Initial Response to Reviewer 34kz (2/2)**
>
> ### **[W2 - Compare against LoRA tuning: Sure! Our methods outperform LoRA in more recent and larger models]**
>
> Below, we compare our approach and LoRA tuning on LLaMA-2-13B and BLOOM-7B models with 3-bit quantization using GPTQ.
>
> > Results suggest that our soft prompt approach vastly outperforms LoRA. In fact, the 3-bit quantized LLaMA-2-13B and BLOOM-7B equipped with our approach may even exceed their FP16 counterparts.
>
> | Dataset | Model                       | Precision | Recover method | Trainable Params (M) | ppl   |
> |---------|-----------------------------|-------|--------|----------------------|-----------|
> | C4      | LLaMA-2-13B                 | fp16  | NA     | NA                    | 6.96      |
> | C4      | LLaMA-2-13B           | 3bit  | NA     | NA                    | 9.24      |
> | C4      | LLaMA-2-13B           | 3bit  | Ours   | 0.5                | **6.75**      |
> | C4      | LLaMA-2-13B           | 3bit  | LoRA   | 26               | 8.15      |
> | C4      | BLOOM-7B                     | fp16  | NA     | NA                    | 15.87     |
> | C4      | BLOOM-7B                     | 3bit  | NA     | NA                    | 18.40     |
> | C4      | BLOOM-7B                     | 3bit  |  Ours  | 0.4               | **13.54**     |
> | C4      | BLOOM-7B                     | 3bit  | LoRA   | 15.7             | 17.26     |
>
>
> ---
>
> Given that both LoRA weights and soft prompts are technically transferrable, we also compare the transferability between the two methods by first training them on the C4 dataset, then transferring and testing on the wikitext2 dataset.
>
> > Results suggest our soft prompt approach still outperforms LoRA in this transferred setting.
>
> | Dataset   | Model       | Precision | Method | Transferred Params (M) | ppl       |
> |-----------|-------------|-------|--------|------------------------|---------------|
> | wikitext2 | LLaMA-2-13B | fp16  | NA     | NA                      | 5.58          |
> | wikitext2 | LLaMA-2-13B | 3bit  | NA     | NA                      | 7.88          |
> | wikitext2 | LLaMA-2-13B | 3bit  | Ours   | 0.5                  | **5.89**      |
> | wikitext2 | LLaMA-2-13B | 3bit  | LoRA   | 26                 | 7.07     |
> | wikitext2 | BLOOM-7B     | fp16  | NA     | NA                      | 13.26         |
> | wikitext2 | BLOOM-7B     | 3bit  | NA     | NA                      | 16.06         |
> | wikitext2 | BLOOM-7B     | 3bit  | Ours   | 0.4                 | **12.42**     |
> | wikitext2 | BLOOM-7B     | 3bit  | LoRA   | 15.7              | 15.65    |
>
>
> ### **[Q1 - Can the prompt prefix generalize across different models? Depending on whether such models share the same tokenizer and embedding dim]**
>
> A learned prompt can be generalized within the models that share the same tokenizer & embedding dim (e.g., LLaMA 1 and 2, or simply any model with different compression levels). While projecting soft prompt into different dimensions is certainly doable (e.g., project LLaMA-7B trained soft prompt with `4096` dim into `8192` to work with LLaMA-65B's), we leave it to future work. Yet, models with different tokenizers will require extra steps to form alignment, as different tokenizers will interpret the same soft prompt differently.

---

### Official Review · Reviewer_QUfQ · 2023-10-30

**Soundness:** 3 good
**Presentation:** 3 good
**Contribution:** 2 fair
**Rating:** 5
**Confidence:** 4

**Summary:**

This paper presents an interesting prompt engineering observation about efficient LLMs. Performance improves by adding a hard-coded prompt telling the LLM to reconsider its solution because it is compressed! The authors build on that observation by performing _transferable_ prompt tuning on a number of compressed (quantized/pruned) LLMs from the OPT/LLAMA family of models.

**Strengths:**

The main benefit claimed by the authors is that the tuned prompts are domain-agnostic. That is, they can be used on different models, datasets, or tasks quite easily because they are related to the fact that the model is compressed, not specifically-tuned for any specific domain.

**Weaknesses:**

The main weaknesses relate to a lack of wider context and evaluation of the presented method. For example: how expensive is prompt tuning? By creating transferable prompts, how much time/effort are we saving? How does accuracy compare to conventional prompt tuning (this is a key missing comparison). How does the presented method peform on other model families? (OPT and Llama are highly related).

Without comparison to other prompt tuning methods, it is hard to put the current results in the needed context.

**Questions:**

please see weaknesses above

---

> ### Author Response · Authors · 2023-11-19
> **Thanks & Initial Response to Reviewer QUfQ (1/2)**
>
> We thank the reviewer for the comments. Below we provide a wider context and evaluation for our proposed method.
>
> ### **[W1.1 - How expensive is prompt tuning? It can be done in hours]**
>
> The backward pass of linear layer, represented as $H^{(l+1)}=H^{(l)}W^{(l)}$, contains two parts, namely, calucating the input gradient $\nabla H^{(l)}=\nabla H^{(l+1)} W^{(l)\top}$ and weight gradient  $\nabla W^{(l)}=H^{(l)\top} \nabla H^{(l+1)}$. **Soft prompt tuning simplifies this process by only requiring the calculation of input gradients.** This approach eliminates the need to cache input tensors, as the model weights are already stored in GPU memory. Consequently, this reduces the backward pass time by 50%, since the weight gradient calculation is bypassed. In contrast to methods like LoRA, which requires caching input tensors for computing the LoRA weight gradient, soft prompt tuning allows larger flexible fine-tuning batch sizes, enhancing GPU utilization. In practice, we can recover the performance of a LLaMA2-7B model within 5 hours on 4 RTX 8000 48G GPUs.
>
> ### **[W1.2 - How much time/effort is saved by transferring prompts? It saves specific prompt tuning efforts and makes models 8X smaller]**
>
> First, regarding time efficiency, transferring prompts across datasets, tasks, and compression levels saves the prompt tuning time for each specific setup. According to our experience with 4 GPUs, that means 5-ish hours per task per compressed model as well as the engineering efforts).
>
> Second, regarding memory usage, as shown in Table 2, our method can recover the accuracy drop of an 8X compressed LLaMA-7B. In practice, this means we reduce the model memory cost from roughly 13GB (in Float16 format) to 1.7GB without sacrificing performance! This substantial decrease in memory usage opens up new opportunities for deploying LLMs on devices with limited resources or inferencing with enlarged batch sizes.
>
>
> ### **[W 1.3 - How does accuracy compare to conventional prompt tuning (this is a key missing comparison): We have already compared them in Appendix A.3. In short, transferred prompts are not as good, but they significantly bridge the gap.]**
>
> If by "conventional prompt tuning," the reviewer means directly learning the soft prompt on the downstream tasks, **then we already have these results in [Table 4 in Appendix A.3](https://openreview.net/pdf?id=Gdm87rRjep#page=14)**. For your convenience, we post our results and summarize our observations here.
>
> > Perplexity comparison between full model and quantized models with different prompts. Where we report test perplexity on PTB and Wikitext-2 dataset. "w./o. prompt" refers to the quantized model without soft prompts. "w./ direct prompt" means the soft prompts are directly trained on the target dataset. "w./ transferred prompt" means the prompt is trained on C4 dataset and then transferred to the target dataset.
>
> | Model                       | PTB     | Wikitext2 |
> |-----------------------------|---------|-----------|
> | Full Model                  | 11.02   | 6.33      |
> | Full Model w./ direct prompt| 6.86    | 5.57      |
> | 4-bit w./o. prompt          | 11.65   | 6.92      |
> | 4-bit w./ direct prompt     | 7.04    | 5.88      |
> | 4-bit w./ transferred prompt| 9.25    | 6.26      |
> | 3-bit w./o. prompt          | 15.74   | 9.45      |
> | 3-bit w./ direct prompt     | 7.76    | 6.33      |
> | 3-bit w./ transferred prompt| 10.81   | 6.90      |
> | 2-bit w./o. prompt          | 5883.13 | 2692.81   |
> | 2-bit w./ direct prompt     | 14.98   | 16.67     |
> | 2-bit w./ transferred prompt| 29.82   | 20.56     |
>
>
> Given direct prompt receives a domain-specific loss, our transferred prompt is, as expected, not as competitive as the direct one. However, such transferred prompt may significantly bridge the gap between a compressed and full model — e.g., our 3-bit & 4-bit quantized LLaMA-7B with transferred prompt can deliver on-par or better PPL than the full model on PTB and Wikitext2. We'd say this is an especially worthy contribution in practice, as one may possibly download the open-sourced transferable prompt to help on a compressed model with little effort.

---

> ### Author Response · Authors · 2023-11-19
> **Thanks & Initial Response to Reviewer QUfQ (2/2)**
>
> ### **[W1.3 Extra - Conventional Prompt tuning cannot transfer between domains!]**
>
> Here we also emphasize that the conventional prompt tuning trained with a domain-specific loss can no longer be transferred between different datasets. Below we present the results of transferring the soft prompts learned on Wikitext2 ( Featured articles on Wikipedia) to PTB (Wall Street Journal material) and C4 (collection of common web text corpus). The results, as shown in the table below, highlight a significant disparity in performance when using domain-specific prompts across different domains. The prompt trained on Wikitext-2, when applied to PTB and C4, leads to a drastic increase in perplexity, indicating a severe degradation in model performance. In contrast, if the prompt is learned on general domain (C4), then it can be transferred to different domain (PTB & Wikitext) and tasks (QA, Commonsense NLI, language understanding, etc. Please check [Appendix A.3](https://openreview.net/pdf?id=Gdm87rRjep#page=15))
>
> > Perplexity comparison on PTB and C4
>
> | Model                       | PTB    | C4    |
> |-----------------------------|---------|---------|
> | Full Model                  | 11.02   |7.59 |
> | 3-bit w./o. prompt          | 15.74   | 10.74|
> | 3-bit w./ prompt learned on Wikitext2          | 294.16   |160.64|
> | 3-bit w./ prompt learned on C4          | 10.81    | 7.48|
>
>
> ### **[W1.4 - Other Model Family: Sure! Our method is still strong on LLaMA-2 and BLOOM]**
>
> We conduct experiments on BLOOM-7B and LLaMA2-13B. Here, we provide the results. We can recover the performance of 3-bit LLaMA2-2-13B and 3-bit BLOOM-7B with even better performance than their fp16 counterparts. The quantization is conducted using GPTQ.
>
> | Dataset | Model                     | Precision | Method | Trainable Params (M) | ppl val  |
> |---------|---------------------------|-------|--------|----------------------|----------|
> | C4      | LLaMA-2-13B | fp16  | NA     | NA                    | 6.96     |
> | C4      | LLaMA-2-13B | 3bit  | NA     | NA                    | 9.24     |
> | C4      | LLaMA-2-13B | 3bit  | Soft Prompt   | 0.5                | **6.75** |
> | C4      | BLOOM-7B                   | fp16  | NA     | NA                    | 15.87    |
> | C4      | BLOOM-7B                   | 3bit  | NA     | NA                    | 18.40    |
> | C4      | BLOOM-7B                   | 3bit  |  Soft Prompt   | 0.4               | **13.54**|
>
> We also test the transferability of our prompt tuning on the wikitext2 dataset. It suggests that our soft prompt can be transferred to other datasets while still outperforming LoRA for the performance recovery of compressed LLMs.
>
> | Dataset   | Model       | Precision | Method | Transferred Params (M) | ppl       |
> |-----------|-------------|-------|--------|------------------------|---------------|
> | wikitext2 | LLaMA-2-13B | fp16  | NA     | NA                      | 5.58          |
> | wikitext2 | LLaMA-2-13B | 3bit  | NA     | NA                      | 7.88          |
> | wikitext2 | LLaMA-2-13B | 3bit  | Ours   | 0.5                  | **5.89**      |
> | wikitext2 | BLOOM-7B     | fp16  | NA     | NA                      | 13.26         |
> | wikitext2 | BLOOM-7B     | 3bit  | NA     | NA                      | 16.06         |
> | wikitext2 | BLOOM-7B     | 3bit  | Ours   | 0.4                 | **12.42**     |

---

> > ### Comment · Reviewer_QUfQ · 2023-11-21
> > **raising my score**
> >
> > The authors addressed many of my main concerns in their rebuttal. The results comparing to other soft prompts and showing non-transferability of conventional methods, in addition to highlighting the cost of prompt tuning makes for a more compelling argument for the presented method. Can I ask the authors to add this content to the main paper? I think it adds more value to the manuscript instead of leaving it buried in the appendix.

---

> > > ### Author Response · Authors · 2023-11-21
> > > **Thank you for raising the score. We are glad to convince you in terms of the evaluation.**
> > >
> > > We thank the reviewer for your valuable time and comments. Your feedback is really helpful in highlighting the unique aspect of our approach compared to the coventential prompt tuning. Although we initially included this comparison experiment in the Appendix, it was not emphasized in the main text. We believe this oversight may be the direct cause of the concerns raised by reviewers `34kz` and `jG4W` regarding our novelty over conventional prompt tuning.
> > >
> > > To address this, we have moved the comparison of our method with conventional prompt tuning into the main text, including experiments which show that the conventional prompt tuning cannot transfer across domains.  We have also explicitly made a point to emphasize this as our key contribution over the previous work in the introduction, as well as the cost comparison between ours and Lora. We also include the detailed fine-tuning time of our method in the experiment section.
> > >
> > >
> > > We hope that the additional adjustment can fully address any remaining issues. Could we kindly inquire if these modifications meet the reviewer's expectations?

---

### Official Review · Reviewer_ARdA · 2023-11-01

**Soundness:** 3 good
**Presentation:** 3 good
**Contribution:** 3 good
**Rating:** 8
**Confidence:** 3

**Summary:**

Motivated by the ability to improve compressed LLM performance through human-engineered prompts (i.e., informing the LLM that it has been compressed), the authors formulate a prompt learning paradigm with the objective of recovering LLM performance lost due to compression/pruning. Additionally, this prompt learned via their method demonstrates transferability between datasets, compression schemes, and tasks. This research is ultimately motivated by the desire for portable, efficient, and accurate LLMs.

**Strengths:**

• Designed and implemented a prompt learning paradigm for improving performance of compressed LLMs through learned prompts.

• Prompt learning paradigm able to recover LLM performance when compared to original model on low and medium quantization/pruning settings. Performance with learned prompt is still better than without in all quantiation/pruning settings.

• Prompt learned for higher pruning/quantization rates is transferrable to lower pruning/quantization rates, respectively. Further, there are some instances where prompt is transferrable from pruning to quantization (or vice versa).

• Prompt learning demonstrated to be compatible with mixed pruned-quantized LLM.

**Weaknesses:**

• Very minor but presentation of some figures could be improved. Consider including baseline value (i.e., value of green line in Figures 2,3,4).

**Questions:**

I do not have any questions. The methodology and presentation was clear to me and the results were easy to interpret. I was going to ask about interpretation of the learned prompts but I found details on that in Appendix D.

---

> ### Author Response · Authors · 2023-11-19
> **Thanks & Initial Response to Reviewer ARdA**
>
> We thank the reviewer for the strong support and meticulous reading!
>
>
> ### **[W1 - Including baseline value in Figures: Sure! Thank you for suggesting it]**
> We have updated the paper by printing out the value for green lines in [Figures 2,3,4](https://openreview.net/pdf?id=Gdm87rRjep#page=4). Please see the revised submission for a better presentation.

---

### Author Response · Authors · 2023-11-19
**Revision Summary**

We thank all the reviewers for their time and effort in helping us improve the quality of the paper. We were glad that the reviewers found the problem interesting and timely (ARdA, 34kz, jG4W). The reviewers also agreed that our experiments on transferring prompts were compelling and extensive (ARdA, 34kz, jG4W).

We have updated the paper to incorporate constructive suggestions. We summarize the major changes:

1. [ARdA] Highlight the full model infernece baseline PPL in [Figures 2,3,4](https://openreview.net/pdf?id=Gdm87rRjep#page=4).
2. [QufQ,34kz] An additional experiments on LLaMA-2-13B and BLOOM-7B in [Appendix A5](https://openreview.net/pdf?id=Gdm87rRjep#page=15).
3. [34kz] An additional experiments on comparing our soft prompts with LoRA LLaMA-2-13b and BLOOM-7B in [Appendix A5](https://openreview.net/pdf?id=Gdm87rRjep#page=15).
4. [QufQ, jG4W] An additional experiments on transferring traditional domain-specific prompts to other domains in the [experiment section](https://openreview.net/pdf?id=Gdm87rRjep#page=8).
5. [QufQ] We move the comparision between our method and traditional prompt tuning to the [experiment section](https://openreview.net/pdf?id=Gdm87rRjep#page=8).
6. [QufQ] We explicitly make a point to emphasize our key contribution over the previous work in the [introduction](https://openreview.net/pdf?id=Gdm87rRjep#page=2), as well as the cost comparison between ours and Lora.

---

### Meta-Review · Area_Chair_831u · 2023-12-07

**Metareview:**

The authors propose and evaluate a prompting-based PEFT method. In particular, they find using a soft-prompting strategy with a compressed model can recover some of the performance lost in compression. The novelty on offer here is slight (insofar that soft prompting is a well-known technique), but it is interesting to see its benefit in the context of compressed models.

However, as pointed out by reviewers, the empirical setup is somewhat limited and makes it difficult to appreciate the scope of the contribution. This was partially addressed in the author response period; integrating these results into the main paper will strengthen the work. So too would making the motivation and analysis connecting prompt tuning and compression tighter; currently, the paper otherwise seems like it is simply applying this technique to compressed models, which may be "novel" in its intersection, but it is not clear why we should expect prompt tuning specifically to help here—making this more explicit would help solidify the contribution.

**Justification For Why Not Higher Score:**

Ultimately, the limited experimental setup and the lack of explicit motivation tying together prompt tuning and model compression weaken the contribution here.

**Justification For Why Not Lower Score:**

N/A

---

### Decision · Program_Chairs · 2024-01-16

Reject